# Honokiol and Alpha-Mangostin Inhibit Mayaro Virus Replication through Different Mechanisms

**DOI:** 10.3390/molecules27217362

**Published:** 2022-10-29

**Authors:** Patricia Valdés-Torres, Dalkiria Campos, Madhvi Bhakta, Paola Elaine Galán-Jurado, Armando A. Durant-Archibold, José González-Santamaría

**Affiliations:** 1Grupo de Biología Celular y Molecular de Arbovirus, Instituto Conmemorativo Gorgas de Estudios de la Salud, Panama City 0816-02593, Panama; 2Programa de Maestría en Microbiología Ambiental, Universidad de Panamá, Panama City 0824-03366, Panama; 3Departamento de Bioquímica, Facultad de Ciencias Naturales, Exactas y Tecnología, Universidad de Panamá, Panama City 0824-03366, Panama

**Keywords:** Honokiol, α-Mangostin, antiviral activity, arboviruses, Mayaro, Una, Chikungunya, Zika, broad-spectrum activity

## Abstract

Mayaro virus (MAYV) is an emerging arbovirus with an increasing circulation across the Americas. In the present study, we evaluated the potential antiviral activity of the following natural compounds against MAYV and other arboviruses: Sanguinarine, (R)-Shikonin, Fisetin, Honokiol, Tanshinone IIA, and α-Mangostin. Sanguinarine and Shikonin showed significant cytotoxicity, whereas Fisetin, Honokiol, Tanshinone IIA, and α-Mangostin were well tolerated in all the cell lines tested. Honokiol and α-Mangostin treatment protected Vero-E6 cells against MAYV-induced damage and resulted in a dose-dependent reduction in viral progeny yields for each of the MAYV strains and human cell lines assessed. These compounds also reduced MAYV viral RNA replication in HeLa cells. In addition, Honokiol and α-Mangostin disrupted MAYV infection at different stages of the virus life cycle. Moreover, Honokiol and α-Mangostin decreased Una, Chikungunya, and Zika viral titers and downmodulated the expression of E1 and nsP1 viral proteins from MAYV, Una, and Chikungunya. Finally, in Honokiol- and α-Mangostin-treated HeLa cells, we observed an upregulation in the expression of type I interferon and specific interferon-stimulated genes, including *IFN**α*, *IFN*β, *MxA*, *ISG15*, *OAS2*, *MDA-5*, *TNF**α*, and IL-1β, which may promote an antiviral cellular state. Our results indicate that Honokiol and α-Mangostin present potential broad-spectrum activity against different arboviruses through different mechanisms.

## 1. Introduction

Arthropod-borne viruses (arboviruses) have provoked large epidemics in the Americas, including the Chikungunya (CHIKV) and Zika (ZIKV) viruses in 2013 and 2015, respectively [1,2]. Endemic and emerging arboviruses, such as Mayaro virus (MAYV), show increasing activity across the region [3,4,5,6,7]. MAYV is a neglected arbovirus belonging to the *Togaviridae* family within the *Alphavirus* genus [8]. MAYV causes Mayaro fever, a disease with non-specific symptoms similar to those of other arboviruses, including fever, headache, diarrhea, leucopenia, retro-orbital pain, myalgia, joint pain, skin rash, and—in some cases—severe polyarthralgia that can last from months to years [9,10]. Although MAYV is mainly transmitted in a sylvatic cycle through the bites of *Haemagogus jantinomys* mosquitoes, recent evidence suggests that urban vectors, such as *Aedes aegypti* or *Aedes albopictus*, may contribute to the spread of this virus, increasing the risk for future outbreaks [11,12,13,14,15]. Despite MAYV’s potential threat to public health, there are currently no licensed vaccines or treatments to combat this infection. Therefore, there is an urgent need to identify potential anti-MAYV drugs.

Natural products are a rich source of molecules with diverse biological activities; in addition, while some natural compounds have been shown to protect against viruses of specific families, others have demonstrated broad-spectrum antiviral activity [16,17,18]. Drugs derived from natural sources have several advantages over synthetic compounds, among them are lower cost, fewer side effects, more diversity, and complexity of chemical molecules—which may limit viral drug resistance and their cost effectiveness in drug discovery programs [19]. All these characteristics suggest that screening natural compounds is a promising strategy for identifying new potential antivirals. Various studies have explored the antiviral effects of natural compounds on MAYV. For example, flavonoids derived from *Bauhinia longifolia*, including Quercetin and Quercetin 3-O-glycosides, were shown to inhibit MAYV replication in a dose-dependent manner [20]. In another study, Ferraz and colleagues found that the flavonoid proanthocyanidin isolated from *Maytenis imbricata* roots demonstrated potent virucidal activity against MAYV [21]. Using a mouse model of MAYV infection, the same research group found that the natural compound, silymarin, prevented liver damage and inflammation, as well as decreased viral load in the liver, spleen, brain, thigh muscle, and footpad [22]. In addition, in our laboratory, Ginkgolic acid isolated from the *Ginkgo biloba* plant showed a strong virucidal activity against Mayaro, Chikungunya, Una, and Zika viruses [23].

Sanguinarine, (R)-Shikonin, Tanshinone IIA, Honokiol, and α-Mangostin are all natural compounds that have been isolated from the plant species *Sanguinaria canadensis*, *Lithospermum erythrorhizon*, *Salvia miltiorrhiza*, *Magnolia officinalis*, and *Garcinia mangostana*, respectively [24,25,26,27,28]. Fisetin is a flavonoid that has been isolated from different plant species in the Fabaceae and Anarcadiaceae families, as well as in several comestible fruits [29]. This compound has demonstrated potent anti-inflammatory, antioxidant, and antitumoral activity [29]. Sanguinarine is a benzophenanthridine alkaloid with anticancer and anti-inflammatory properties [24,30,31]. Shikonin is a naphthoquinone with potential application for the treatment of several types of tumors, inflammation, and wound healing [25,32]. Tanshinone IIA is a diterpene quinone with antioxidant and anti-inflammatory properties [27]. Honokiol is a lignan biphenol, whereas α-Mangostin belongs to a class of molecules know as xanthones. These compounds have all shown multiple pharmacological activities, including anti-inflammatory [33,34,35], antifungal [36,37], antifibrotic [38,39], antibacterial [40,41,42], antitumoral [43,44,45,46], antioxidant [47,48], anti-depressant [26,49], neuro- and cardio-protective properties [50,51,52,53], as well as antiviral activity against certain viruses [54,55,56,57,58]. However, neither Sanguinarine, (R)-Shikonin, Fisetin, Tanshinone IIA, Honokiol, nor α-Mangostin have been studied extensively in the context of arboviruses. Thus, the aim of this work was to evaluate the potential antiviral activity of these natural compounds against MAYV and other arboviruses. 

## 2. Results

### 2.1. Honokiol and α-Mangostin Prevent MAYV-Induced Cytopathic Effects in Vero-E6 Cells in a Dose-Dependent Manner

To explore the potential antiviral activity of the natural compounds Sanguinarine, (R)-Shikonin, Fisetin, Honokiol, Tanshinone IIA, and α-Mangostin (Figure 1A–F) against MAYV, we first analyzed the cytotoxicity of these compounds in Vero-E6 cells using the MTT method. As shown in Figure 2, Sanguinarine and (R)-Shikonin significantly reduced cell viability at doses of 5 and 10 μM for both incubation times tested (Figure 2A,B). In contrast, with Fisetin, Honokiol, and Tanshinone IIA, cell viability was around 80% or higher, independent of the incubation time (Figure 2C–E). In the case of α-Mangostin, we observed a high toxicity at 10 μM concentration for both incubation times tested, while the 5 μM dose appeared to have no effect (Figure 2F). Thus, we decided not to include Sanguinarine and (R)-Shikonin in further experiments, and we used 10 μM for Fisetin, Honokiol, and Tanshinone IIA, as well as 5 μM for α-Mangostin, as the maximal doses in the subsequent experiments.

Previous studies have determined that MAYV has the capacity to induce strong cytopathic effects in different cell lines, including in Vero and primary human dermal fibroblasts (HDFs) [59,60]. Therefore, we evaluated whether the Fisetin, Honokiol, Tanshinone IIA, or α-Mangostin compounds were able to protect Vero-E6 cells from MAYV-induced damage. Microscopic analysis of MAYV-infected cells revealed that Fisetin did not appear to protect cells from virus-induced cytopathic effects, regardless of the dose tested (Figure 3), while we observed a partial protection when using the higher concentration of Tanshinone IIA that was evaluated (Figure 3). On the other hand, cell protection was evident with Honokiol and α-Mangostin, with the higher dose offering greater protection (Figure 3). Taken together, these results indicate that Honokiol and α-Mangostin block MAYV-induced cytopathic effects in Vero-E6 cells and suggest that these natural compounds may have antiviral activity.

### 2.2. Honokiol and α-Mangostin Reduce MAYV Replication in Vero-E6 Cells in a Dose-Dependent Manner

To investigate if Fisetin, Honokiol, Tanshinone IIA, or α-Mangostin influence MAYV replication, we assessed viral progeny production in supernatants from infected Vero-E6 cells treated with increasing concentrations of these compounds using plaque-forming assays. While Fisetin did not affect MAYV progeny production, Tanshinone IIA resulted in a small but significant reduction in viral titers (Figure 4A,C). In contrast, Honokiol and α-Mangostin promoted a substantial dose-dependent decrease in viral titers (Figure 4B,D), which reached between 3 and 4 logs at the maximum doses tested (Figure 4B,D). To corroborate these findings, we performed a similar experiment and used an immunofluorescence assay in order to analyze the percentage of E1 protein-positive cells among MAYV infected-cells treated with Honokiol or α-Mangostin. In MAYV-infected cells treated with DMSO, 64.9 ± 6.8% of Vero-E6 cells were positive for E1 protein (Figure 4E,F). Interestingly, in Vero-E6 cells treated with Honokiol or α-Mangostin we found a significant decrease in MAYV-infected cells (37.3 ± 10.7% and 10.8 ± 4.6% for Honokiol; 42.9 ± 8% and 18.5 ± 7.3% for α-Mangostin), and this effect was dose-dependent (Figure 4E,F). These results confirm that Honokiol and α-Mangostin inhibit MAYV replication in Vero-E6 cells.

### 2.3. Honokiol and α-Mangostin Inhibit MAYV Progeny Production Independent of Virus Strain or Human Cell Line Tested

Up until this point, the infection experiments were performed using the MAYV strain AVR0565 that was isolated in San Martin, Peru. To determine if the antiviral effect of Honokiol and/or α-Mangostin is also present in other MAYV strains from different geographical areas, we tested both compounds on the Guyane (Guyane, French Guiana) and TRVL4675 (Mayaro, Trinidad and Tobago) strains. To this end, we infected Vero-E6 cells with these strains and then treated them with increasing doses of Honokiol or α-Mangostin. Following 24 h of incubation, we quantified MAYV progeny production. This analysis revealed that Honokiol and α-Mangostin reduced MAYV progeny yield regardless of the virus strains being tested (Figure 5A–D). To validate these results, we used two human cell lines, primary HDFs, and HeLa cells, which were previously demonstrated to be susceptible to MAYV infection [60]. We infected both cell lines with the MAYV strain AVR0565 and applied Honokiol or α-Mangostin, as described above. Again, we found that Honokiol and α-Mangostin decreased viral progeny production regardless of the cell line tested (Figure 5E–H). The inhibitory effect of these compounds on the human cell lines we evaluated did not appear to be associated with cell toxicity (Figure 5I–L). These results provide further evidence that Honokiol and α-Mangostin inhibit MAYV replication.

### 2.4. Pretreating HDFs with α-Mangostin, but Not Honokiol, Affects MAYV Progeny Production

While in all the preceding assays the Honokiol or α-Mangostin treatment was applied after viral adsorption, we decided instead to evaluate whether pretreating HDFs with these compounds has any effect on MAYV replication. To this end, HDFs were pre-treated with Honokiol or α-Mangostin for 2 h and then infected with MAYV as previously described. After viral adsorption, a fresh medium without the compounds was added to the cells. They were then incubated for an additional 24 h and viral titers were assessed as described above. As shown in Figure 6, in HDFs pre-treated with α-Mangostin there was a significant dose-dependent reduction in viral titers, whereas with Honokiol, we did not observe any effect (Figure 6A,B). In order to determine if Honokiol or α-Mangostin act directly on MAYV particles, we performed a virucidal assay. In these experiments, we did not observe a decrease in viral titers in the solutions containing Honokiol or α-Mangostin when compared to control solutions prepared with DMSO, indicating that these compounds do not have a direct effect on MAYV (Figure 6C,D).

### 2.5. Honokiol and α-Mangostin Treatment Disturb MAYV Infection at Different Stages of the Viral Life Cycle

In an attempt to identify the MAYV cycle stage and how it is affected by Honokiol or α-Mangostin, we treated cells with these compounds at different phases of virus infection. The first step in MAYV infection consists of viral particles attaching to a receptor on the cell membrane of a susceptible host cell [8]. Thus, we completed a binding assay in which HDFs were infected with MAYV in the presence of Honokiol or α-Mangostin at 4 °C for 1 h. At this temperature, the virus is able to attach to cell membrane receptors, but not enter the cells. Then, the cells were incubated at 37 °C in the medium without the compounds for 24 h and the viral titers were then quantified. These experiments revealed that α-Mangostin affected MAYV attachment for both of the doses tested (Figure 7B). In Honokiol-treated cells, we observed a modest effect at the higher concentration we tested (Figure 7A). Next, we wanted to evaluate whether Honokiol or α-Mangostin may affect viral entry into the host cell. To achieve this, we infected HDFs with MAYV at 4 °C; after 1 h, the cells were shifted to 37 °C and incubated with Honokiol or α-Mangostin for 1 h. Then, the compounds were removed and, following 24 h of infection, viral progeny production was evaluated. These assays demonstrated that α-Mangostin partially disturbed the viral entry step, while Honokiol did not appear to affect this process (Figure 7C,D). Finally, we carried out a post-entry assay. We infected the cells using the same procedure as the entry assay, where, after viral adsorption, we incubated the cells at 37 °C for 2 h. Next, we added Honokiol or α-Mangostin, the cells were then incubated until 24 h post infection and viral titers were assessed as described above. As shown in Figure 7E,F both natural compounds were able to reduce viral progeny production, indicating that Honokiol and α-Mangostin also affect a post-entry step in MAYV infection. Collectively, these data suggest that these compounds inhibit MAYV infection at diverse stages of the MAYV life cycle.

### 2.6. Honokiol and α-Mangostin Downmodulate the Expression of MAYV E1 and nsP1 Proteins and Additionally, Affect Viral RNA Replication

To examine the effect of Honokiol or α-Mangostin on the expression of the MAYV E1 and nsP1 proteins, we completed an infection experiment in cells treated with increasing doses of Honokiol or α-Mangostin and analyzed the results using Western blot. As shown in Figure 8, Honokiol and α-Mangostin promoted a significant dose-dependent decrease in both viral proteins in HeLa cells (Figure 8A,B). Similar results were observed in Vero-E6 cells and HDFs treated with these compounds (Appendix A). To evaluate if these compounds affect viral RNA replication, we performed a RT-PCR in HeLa cells infected with MAYV and treated with Honokiol or α-Mangostin. Our results indicated that treatment with these compounds triggered a potent reduction in viral RNA. Collectively, these results indicate that Honokiol and α-Mangostin affect the expression of MAYV E1 and nsP1 proteins as well as viral RNA replication.

### 2.7. Honokiol and α-Mangostin Also Inhibit the Una, Chikungunya, and Zika Arboviruses

Given that Honokiol and α-Mangostin have demonstrated significant inhibitory activity with MAYV, we decided to evaluate the effect of these compounds on other emerging and re-emerging arboviruses. For this, Vero-E6 cells were infected with the alphaviruses Una (UNAV), Chikungunya (CHIKV), or the Flavivirus Zika (ZIKV); then, increasing doses of Honokiol or α-Mangostin in fresh medium were added to the cells after 1 h of virus adsorption. Following 24 h of incubation, viral titers and the expression of the E1 and nsP1 proteins were measured in cell supernatants and lysates, respectively. Our results indicate that the treatment with Honokiol or α-Mangostin promoted a substantial reduction in viral titers for all the arboviruses tested (Figure 9A,B,E,F,I,J). Moreover, we observed a downmodulation of E1 and nsP1 protein expression in UNAV- and CHIKV-infected cells treated with these compounds (Figure 9C,D,G,H). Unfortunately, two commercial antibodies against the ZIKV NS1 and E proteins did not function as expected in our experiments. These findings suggest that Honokiol and α-Mangostin may have broad-spectrum antiviral activity.

### 2.8. Honokiol and α-Mangostin Treatment Elicit the Expression of Type I Interferon and Specific Interferon-Stimulated Genes in HeLa Cells

In order to evaluate another possible mechanism by which Honokiol and α-Mangostin inhibit viral replication, we explored the interferon pathway. Type I interferon (IFN) production is one of the main arms of cell defense used to control viruses and other microbial infections [61]. Type I IFN includes five proteins: IFNα, IFNβ, IFNκ, IFNε, and IFNω. IFNα/β induces the JAK–STAT pathway, which promotes the expression of antiviral genes implicated in the immune response [61]. Previously, Chen and collaborators showed that Honokiol, and its isomere Magnolol, were able to induce type I IFN and IFN-stimulated genes in carp kidney cells, contributing to grass carp reovirus inhibition [62]. Moreover, α-Mangostin has been demonstrated to stimulate IFN synthesis through the TBK1-IRF-3 pathway and act as an agonist of adaptor protein stimulator of interferon genes (STING) in human macrophages [63]. To investigate if the antiviral activity observed for Honokiol or α-Mangostin in HeLa and HDFs may be mediated by IFN in these cell infection models, we assessed the expression of type I IFN and specific IFN-stimulated genes in treated or untreated HeLa cells using quantitative RT-PCR. In these experiments, we found that Honokiol or α-Mangostin treatment promoted a significant increase in mRNA expression for the IFNα, IFNβ, MxA, ISG15, OAS2, and MDA-5 genes (Figure 10A–F). In addition, Honokiol stimulated the expression of inflammatory cytokine genes, such as TNFα and IL-1β (Figure 10G,H). Taken together, these findings indicate that Honokiol and α-Mangostin may elicit an antiviral cellular state through a possible modulation of the IFN pathway in HeLa cells.

## 3. Discussion

MAYV is a neglected and emerging arbovirus with increasing activity across the Americas [3]. Although MAYV is mainly transmitted by sylvatic mosquito species in tropical regions, growing evidence indicates that urban vectors, such as *Aedes aegypti* or *Aedes albopictus*, may contribute to the spread of this pathogen, increasing the risk for future epidemics [13]. Despite MAYV’s potential threat to public health, there are no approved drugs to combat this virus. Therefore, identifying potential anti-MAYV treatments remains crucial.

Natural products are a common source of molecules with a broad range of pharmacological activities, including antiviral compounds [16,17]. In the present study, we used a series of in vitro assays to investigate the potential antiviral activity of plant-derived compounds against MAYV and other arboviruses. The compounds tested included Sanguinarine chloride, (R)-Shikonin, Fisetin, Honokiol, Tanshinone IIA, and α-Mangostin. Cytotoxicity analysis revealed that Sanguinarine chloride and (R)-Shikonin had considerable toxic effects for all doses and incubation times tested. These findings are consistent with previous reports of these compounds’ high cytotoxicity in several cancer cell lines, supporting their possible use as antitumoral drugs [30,31,32,64]. On the other hand, Fisetin, Honokiol, Tanshinone IIA, and α-Mangostin were well-tolerated at doses of 5 or 10 μM in the tested cell lines. Therefore, we evaluated their ability to protect Vero-E6 cells from MAYV-induced cytopathic effects. In these experiments, we found that Honokiol and α-Mangostin prevented MAYV-induced damage in a dose-dependent manner, indicating these compounds may have antiviral activity. In contrast, Fisetin did not protect the Vero-E6 cells, and Tanshinone IIA showed a slight protection only at the highest concentration tested.

To further explore our hypothesis, we assessed MAYV progeny production in Vero-E6 cells treated with increasing doses of Fisetin, Honokiol, Tanshinone IIA, or α-Mangostin. Our results demonstrated that Honokiol and α-Mangostin promoted a significant dose-dependent reduction in MAYV viral titers. This decrease reached between 3 and 4 logs at the maximum dose tested. In agreement with our observations for the cell protection assay, we did not find a decline in viral titers in Fisetin-treated cells, and we saw only a modest effect with Tanshinone IIA treatment. In addition, we analyzed the percentage of MAYV E1 protein-positive cells in Honokiol- and α-Mangostin-treated cells using an immunofluorescence assay. These experiments indicated that Honokiol and α-Mangostin reduced the percentage of MAYV-infected cells in a dose-dependent manner. To further validate these findings, we tested the effects of Honokiol and α-Mangostin with two additional MAYV strains, Guyane and TRVL4675, and using two human cell lines, HDFs and HeLa. We obtained similar results again, providing further evidence that these compounds exhibit anti-MAYV activity. The antiviral activity we observed is in line with previous studies, which have revealed that Honokiol or α-Mangostin inhibit Dengue virus serotype-2, CHIKV, human norovirus, Herpes simplex virus-1, hepatitis C virus, and grass carp reovirus [54,55,56,57,58,62,65]. 

The previously described experiments from our study involved applying Honokiol or α-Mangostin after viral adsorption, but we also explored the consequences of pretreating cells with these compounds. In these assays we found that only pretreatment with α-Mangostin affected MAYV progeny production. We also evaluated the possible effects of Honokiol or α-Mangostin on MAYV particles. However, the virucidal assay results indicate that Honokiol and α-Mangostin did not act directly on MAYV. To identify the stage of the MAYV viral cycle impacted by Honokiol or α-Mangostin, we completed binding, entry, and post-entry assays. Our findings revealed that α-Mangostin treatment led to effects on all three of these viral stages, whereas Honokiol treatment only partially affected the viral binding and post-entry stages, which indicates these natural compounds may block MAYV infection at distinct phases. Consequently, we assessed the expression of E1 and nsP1 viral proteins in Honokiol- or α-Mangostin-treated cells. Western blot analyses demonstrated that Honokiol and α-Mangostin promoted a downmodulation of both viral proteins for all the cell lines tested. In addition, RT-PCR experiments in HeLa cells revealed that these compounds strongly suppressed viral RNA replication. Collectively, these findings suggest that Honokiol and α-Mangostin may inhibit viral replication mainly through two mechanisms: the suppression of viral proteins expression and decreased viral RNA replication. Earlier work has demonstrated that Honokiol affects the Dengue virus entry step by blocking the endocytic pathway and reduces the expression of NS1 and NS3 viral proteins [54]. 

Since Honokiol and α-Mangostin have shown consistent suppressive activity against MAYV, we decided to examine the effects of these compounds on other arboviruses: UNAV, CHIKV, and ZIKV. Although a previous study reported that α-Mangostin disrupted the replication of an African genotype strain of CHIKV in vitro and in vivo [58], we tested this compound on a CHIKV strain (with an Asian lineage) that was isolated in Panama [66]. Viral titer quantification experiments showed that Honokiol and α-Mangostin decreased viral progeny production in a dose-dependent manner for all the arboviruses we assessed. Moreover, we observed a significant reduction in E1 and nsP1 protein levels in cell lysates from UNAV- and CHIKV-infected cells, indicating that these compounds may have broad-spectrum antiviral activity. It is important to highlight this is the first report of Honokiol’s antiviral activity against different alphaviruses. 

Previous studies have shown that Honokiol and α-Mangostin were able to modulate antiviral cell defense by activating the interferon pathway. Chen and collaborators found that Honokiol and a related compound, Magnolol, enhanced the antiviral response against grass carp reovirus via increased transcription of type I IFN and IFN-stimulated genes in carp kidney cells [62]. Furthermore, Zhang et al. demonstrated that α-Mangostin activates the adaptor protein STING in human macrophages, thus promoting type I IFN synthesis [63]. While our preceding experiments in Vero-E6 cells indicated that Honokiol and α-Mangostin’s antiviral activity is independent of the interferon pathway, we decided to evaluate the expression of type I IFN and IFN-stimulated genes in Honokiol- or α-Mangostin-treated HeLa cells in order to explain our observations. These assays showed that both compounds upregulated the expression of the *IFN**α*, *IFNβ*, *MxA*, *ISG15*, *OAS2*, and *MDA-5* genes. Honokiol was also able to induce the expression of *TNF**α* and *IL-1β* genes. Collectively, these findings suggest that Honokiol and α-Mangostin may also inhibit viral replication through a possible modulation of the IFN pathway, at least in Hela cells. 

Although Honokiol and α-Mangostin have a limited oral bioavailability—thereby affecting their potential use as antivirals—several delivery systems, including nanoparticles or nanomicelles, have been developed in order to improve the activity, bioavailability, and pharmacokinetic properties of these natural compounds [67,68]. An alternative strategy to be explored is an analogue compound synthesis, which could provide similar or enhanced activities, but with improved pharmacological profiles [69,70]. Our results support that Honokiol and α-Mangostin compounds represent potential broad-spectrum antivirals that act through different mechanisms. However, detailed studies in animal models are necessary to determine the utility and efficacy of these compounds as antiviral drugs.

## 4. Materials and Methods

### 4.1. Cell Culture and Reagents

Vero-E6 cells (CRL-1586), human dermal fibroblasts (HDFs) from adults (PCS-201-012) (both obtained from ATCC, Manassas, VA, USA) and HeLa cells (kindly provided by Dr. Carmen Rivas, CIMUS, Santiago de Compostela, Spain) were grown in a Minimal Essential Medium (MEM) or in a Dulbecco’s Modified Eagle’s Medium (DMEM) supplemented with a 10% fetal bovine serum (FBS), a 1% penicillin-streptomycin antibiotic solution, and 2 mM of _L_-Glutamine (all reagents were obtained from Gibco, Waltham, MA, USA). Cell lines were incubated at 37 °C under a 5% CO_2_ atmosphere. The natural compounds Sanguinarine chloride (13-methyl-[1,3]-benzodioxolo [5,6-*c*]-1,3-dioxolo[4,5-*i*]phenanthridinium chloride), (97.8% purity); (R)-Shikonin (5,8-dihydroxy-2-[(1*R*)-1-hydroxy-4-methyl-3-penten-1-yl]-1,4-naphthalenedione), (99.8% purity); Fisetin (2-(3,4-Dihydroxyphenyl)-3,7-dihydroxy-4*H*-1-benzopyran-4-one), (98.0% purity); Honokiol (5,3′-Diallyl-2,4′-dihydroxybiphenyl), (99.9% purity); Tanshinone IIA (6,7,8,9-Tetrahydro-1,6,6-trimethylphenanthro[1,2-*b*]furan-10,11-dione), (98.6% purity); and α-Mangostin (1,3,6-Trihydroxy-7-methoxy-2,8-bis(3-methyl-2-buten-1-yl)-9*H*-xanthen-9-one), (97.7% purity), were all obtained from Tocris (Minneapolis, MN, USA). All compounds were dissolved in Dimethyl sulfoxide (DMSO, Sigma-Aldrich, St. Louis, MI, USA) at 10 mM concentration and stored at −20 °C until use. Working solutions of the natural compounds were prepared in MEM or DMEM at the indicated concentrations.

### 4.2. Virus Strains and Propagation

The Mayaro (MAYV, AVR0565, San Martín, Peru; MAYV, Guyane, Guyane, French Gianna; and MAYV, TRVL4675, Mayaro, Trinidad and Tobago) and Una (UNAV, BT-1495-3, Bocas del Toro, Panama) [71] strains, used in this study, were obtained from the World Reference Center for Emerging Viruses and Arboviruses (WRCEVA) at University of Texas Medical Branch (UTMB), USA and were kindly provided by Dr. Scott Weaver. The Chikungunya (CHIKV, Panama_256137_2014) [66] and Zika (ZIKV, 259249) strains were isolated from patient sera collected during the Chikungunya and Zika epidemics in Panama in 2014 and 2015, respectively. Viruses were propagated in Vero-E6 cells and then titrated, aliquoted, and stored as previously described [23].

### 4.3. Analysis of Cell Toxicity

Cytotoxicity of the natural compounds Sanguinarine chloride, (R)-Shikonin, Fisetin, Honokiol, Tanshinone IIA, and α-Mangostin was evaluated using the MTT method, as previously reported [60]. Briefly, confluent Vero-E6, HDFs, or HeLa cells grown in 96-well plates were treated with the indicated concentrations of each compound or DMSO (0.1%), as a control. After 24 or 48 h of incubation, 5 mg/mL of 3-(4,5-Dimethyl-2-thiazolyl)-2,5-diphenyltetrazolium bromide (MTT, Sigma-Aldrich, St. Louis, MI, USA) solution was applied to the cells and incubated for an additional 4 h. Formazan crystals were dissolved in a solution of 4 mM HCl and 10% Triton X-100 in isopropanol, and absorbance was determined at 570 nm using a microplate reader spectrophotometer (BioTeK, Winooski, VT, USA). Results are shown as the percentage of viable cells relative to untreated control cells.

### 4.4. Plaque-Forming Assay

Viral progeny production in cell supernatants from Vero-E6, HDFs, or HeLa cells infected with MAYV, UNAV, CHIKV, or ZIKV was quantified using plaque-forming assays as previously described [23]. Briefly, 10-fold serial dilutions of infected samples were used to infect confluent Vero-E6 cells grown in 6-well plates. After 1 h of virus adsorption, the inoculum was removed and the cells were overlaid with a solution of 1% agar supplemented with 2% FBS, then incubated for 3 days at 37 °C. Next, the agar was eliminated, and the cells were fixed with 4% formaldehyde solution in PBS and stained with 2% crystal violet dissolved in 30% methanol solution. Finally, the numbers of plaques were calculated, and the viral titers were reported as plaque-forming units per milliliter (PFU/mL).

### 4.5. Viral Infection Assay

Vero-E6, HDFs, or HeLa cells grown in 12- or 24-well plates were infected with MAYV, UNAV, CHIKV, or ZIKV at an MOI of 1. After 1 h of virus adsorption, cells were treated with the indicated doses of the natural compounds, and they were incubated for 24 h. Then, viral titers in cell supernatants were quantified using a plaque-forming assay. For the pretreatment assay, HDFs were pretreated with indicated concentrations of Honokiol or α-Mangostin for 2 h. Then, the compounds were removed, and the cells were infected with MAYV as mentioned above. After that, the cells were incubated for 24 h without the compounds and viral progeny production was quantified. For the binding, entry and post-entry assays the infection was performed at 4 °C. In the binding assay, the infection was carried out in the presence of Honokiol or α-Mangostin. Then, the compounds were eliminated, and the cells were incubated at 37 °C for 24 h, before the viral titers were quantified as previously described. For the entry assay, following 1 h of virus adsorption, cells were shifted to 37 °C, treated with the compounds for 1 h and then incubated for 24 h before evaluating the viral progeny production. Finally, in the post-entry assay, Honokiol or α-Mangostin was added to the cells after 2 h of virus adsorption, incubated for 24 h, and then viral titers were measured using a plaque-forming assay.

### 4.6. Immunofluorescence Assay

Vero-E6 cells grown on glass coverslips were infected with MAYV and then treated with increasing doses of Honokiol or α-Mangostin as indicated above. Following 24 h of infection, cells were fixed, blocked, and permeabilized as previously performed [60]. Next, the cells were stained with a rabbit MAYV E1 antibody, which was previously validated in our laboratory [72], followed by the application of an Alexa Flour 568 goat antirabbit secondary antibody (Invitrogen, Carlsbad, CA, USA). Lastly, coverslips were mounted on slides with Prolong Diamond Antifade Mountant with DAPI in order to stain the cell nuclei (Invitrogen, Carlsbad, CA, USA); further, microphotographs were obtained with an FV1000 Flouview confocal microscope (Olympus, Lombard, IL, USA). Moreover, the images were analyzed with ImageJ software (National Institute of Health, Bethesda, USA).

### 4.7. Western Blot Assay

Vero-E6, HDFs, or HeLa cells were infected with MAYV, UNAV, or CHIKV and after 1 h of virus adsorption, cells were treated with the indicated concentrations of Honokiol or α-Mangostin. Following 24 h of infection, protein extracts were obtained and separated in SDS-PAGE, transferred to nitrocellulose membranes, and blocked with a solution of 5% non-fat milk in T-TBS buffer. Next, membranes were incubated overnight at 4 °C with the following primary antibodies: rabbit polyclonal anti-E1, rabbit polyclonal anti-nsP1 (against alphaviruses)—both previously validated in the laboratory [72]—and mouse monoclonal anti-GAPDH (Cat. # VMA00046, Bio-Rad, Hercules, CA, USA). Afterward, the membranes were washed 3 times with T-TBS buffer and incubated with HRP-conjugated goat anti-rabbit (Cat. # 926-80011) or goat anti-mouse (Cat. # 926-80010) secondary antibodies (LI-COR, Lincoln, NE, USA) for 1 h at room temperature. Lastly, the membranes were incubated with SignalFire^TM^ ECL Reagent (Cell Signaling Technology, Danvers, MA, USA) for 5 min, and the chemiluminescent signal was detected with a C-Digit scanner (LI-COR, Lincoln, NE, USA).

### 4.8. Gene Expression and Viral RNA Analysis by Quantitative RT-PCR

Total RNA was extracted from HeLa cells treated or untreated with Honokiol or α-Mangostin and infected with MAYV using an RNeasy kit (QIAGEN, Valencia, CA, USA) following the manufacturer’s instructions. Single-stranded cDNA was synthetized from 1 μg of RNA using a High-Capacity cDNA Reverse Transcription kit, and quantitative RT-PCR was completed using Power SYBR Green PCR Master Mix in a QuantiStudio^TM^ 5 thermocycler (Applied Biosystems, Foster City, CA, USA) in order to evaluate the mRNA levels of the following genes using the primers listed in Table 1: *IFN**α*, *IFNβ*, *MxA*, *ISG15*, *OAS2*, *MDA-5*, *TNF**α*, and *IL-1β*. In addition, specific primers to detect MAYV were used. Relative mRNA expression was measured using the *β*-actin gene for normalization according to the ∆∆ CT method [73].

### 4.9. Data Analysis

All Experiments were performed at least 3 times in triplicate, unless otherwise stated. For each experiment, the mean ± standard deviation is shown. All data were analyzed using the Mann–Whitney test or one-way ANOVA test followed by Dunnett’s post hoc test. Data analysis was performed, and graphics were created, using GraphPad Prism software version 9.4.1 (GraphPad Software, San Diego, CA, USA) for Mac.

## Figures and Tables

**Figure 1 molecules-27-07362-f001:**
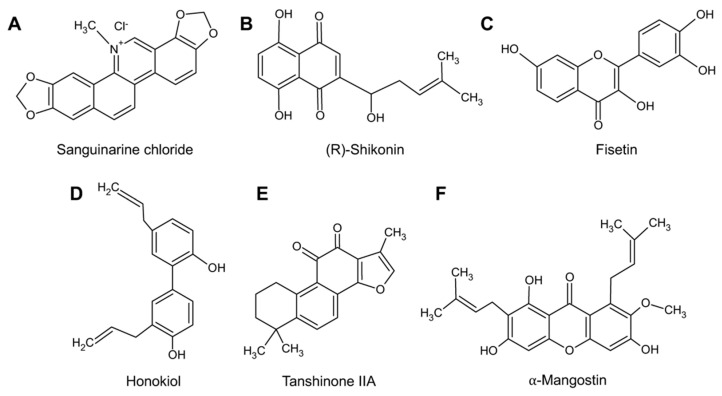
Chemical structures of the natural compounds tested. (**A**) Sanguinarine chloride; (**B**) (R)-Shikonin; (**C**) Fisetin; (**D**) Honokiol; (**E**) Tanshinone IIA; and (**F**) α-Mangostin.

**Figure 2 molecules-27-07362-f002:**
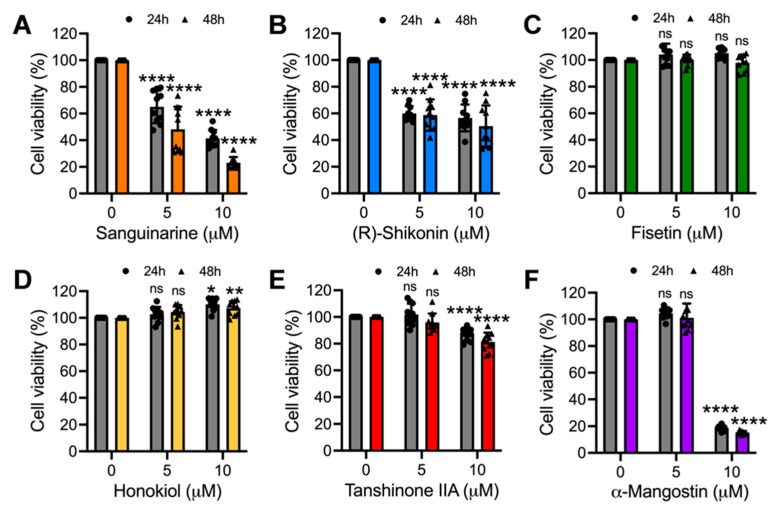
Cytotoxicity of the natural compounds evaluated in this study. Vero-E6 cells were treated with the indicated concentrations of Sanguinarine chloride (**A**), (R)-Shikonin (**B**), Fisetin (**C**), Honokiol (**D**), Tanshinone IIA, (**E**) or α-Mangostin (**F**). After 24 or 48 h of incubation, cell viability was determined using an MTT assay. Data represent the mean ± standard deviation of two independent experiments with five replicates. Data were analyzed with a one-way ANOVA test followed by Dunnett’s post hoc test. Statistically significant differences are denoted as follows: * *p* < 0.05; ** *p* < 0.01; **** *p* < 0.0001; and ns: non-significant.

**Figure 3 molecules-27-07362-f003:**
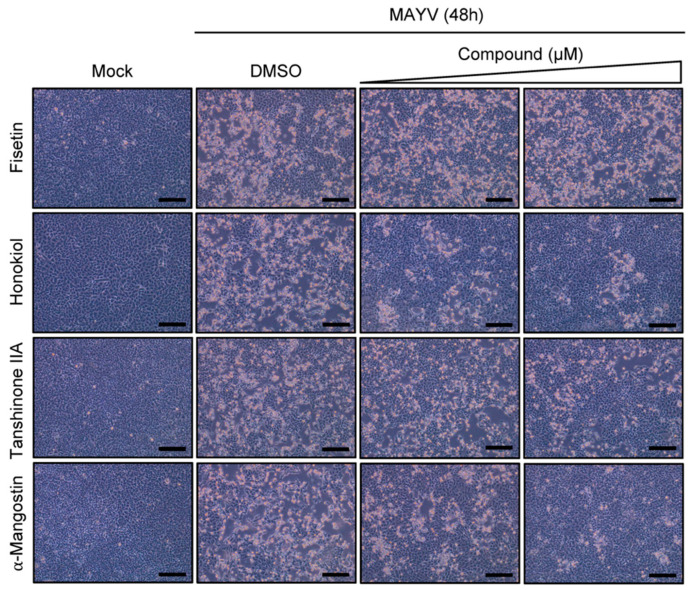
Inhibition of MAYV-induced cytopathic effects by Honokiol and α-Mangostin in Vero-E6 cells is dose-dependent. Vero-E6 cells were infected with the MAYV strain AVR0565 at a multiplicity of infection (MOI) of 1. Further, after 1 h of virus adsorption, cells were treated with Fisetin, Honokiol, Tanshinone IIA (at doses of 5 or 10 μM), or α-Mangostin (at doses of 1 or 5 μM) for 48 h. DMSO (0.1%) served as a control. Cytopathic effects were evaluated using an inverted microscope. One representative microphotograph of at least 10 different fields is shown. Scale bar: 100 μm.

**Figure 4 molecules-27-07362-f004:**
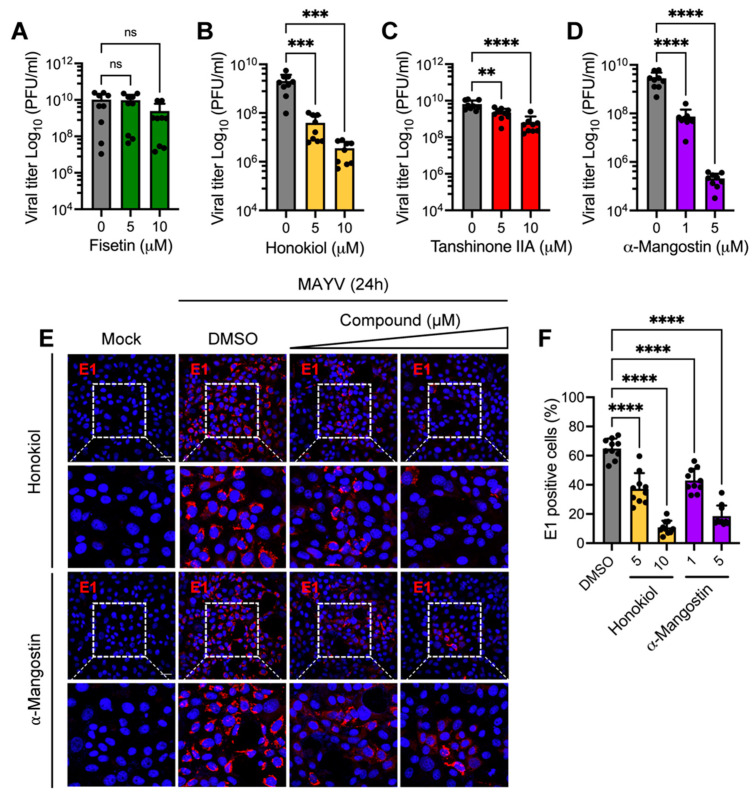
Honokiol and α-Mangostin promote a reduction in MAYV progeny production. Vero-E6 cells were infected with the MAYV strain AVR0565 using an MOI of 1. After 1 h of virus adsorption, cells were treated with the indicated doses of Fisetin (**A**), Honokiol (**B**), Tanshinone IIA (**C**), or α-Mangostin (**D**); further, DMSO (0.1%) was used as a control. After 24 h of incubation, viral progeny production in cell supernatants was quantified using a plaque-forming assay. Data represent the mean ± standard deviation of three independent experiments in triplicate. (**E**) Vero-E6 cells grown on glass coverslips were infected with MAYV and treated with Honokiol or α-Mangostin as indicated above. After 24 h of infection, cells were stained with an MAYV E1 antibody followed by a secondary antibody Alexa-Flour 568 and nuclei were stained with DAPI. Then, the cells were analyzed with an immunofluorescence confocal microscope, with a scale bar of: 30 μm. (**F**) The percentage of MAYV E1 protein-positive cells was determined in at least 10 different fields. Data were analyzed using a one-way ANOVA test followed by Dunnett post hoc test. Statistically significant differences are denoted as follows: ** *p* < 0.01; *** *p* < 0.001; **** *p* < 0.0001; and ns: non-significant.

**Figure 5 molecules-27-07362-f005:**
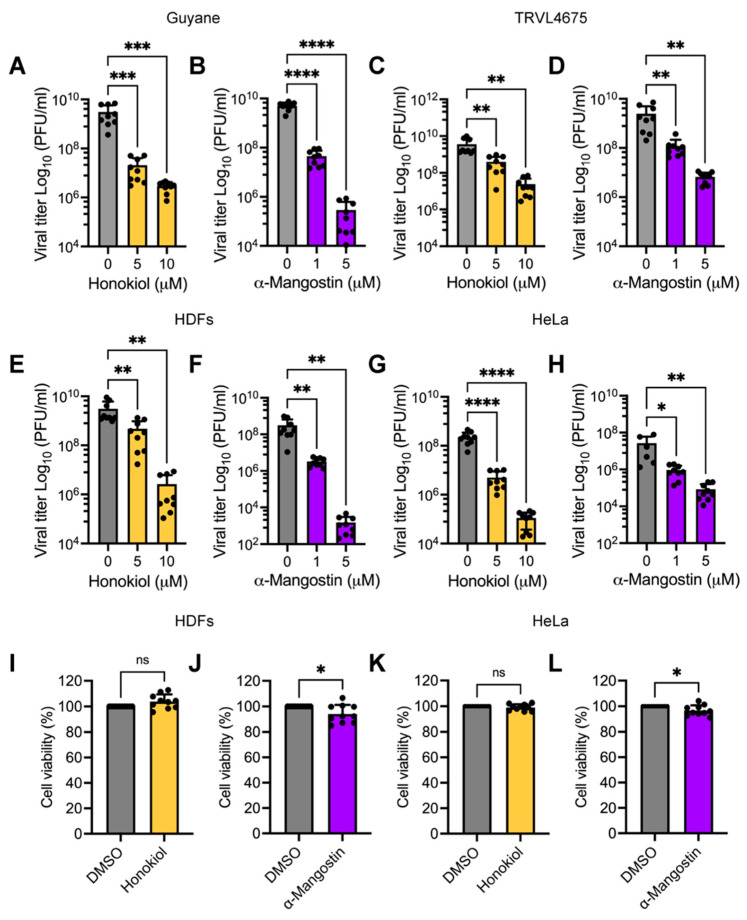
Honokiol and α-Mangostin reduce MAYV progeny yields, regardless of the virus strain or human cell line tested. Vero-E6 cells were infected with the MAYV Guyane (**A**,**B**) or TRVL4675 (**C**,**D**) strains and then treated with Honokiol or α-Mangostin at the indicated concentrations for 24 h. After that, viral progeny production in cell supernatants was quantified using a plaque-forming assay. HDFs (**E**,**F**) or HeLa (**G**,**H**) cells were infected with the MAYV AVR0565 strain and treated as previously indicated. Following 24 h of incubation, viral titers in cell supernatants were analyzed as previously described. Cell viability in HDFs (**I**,**J**) or HeLa cells (**K**,**L**) treated with Honokiol (10 μM) or α-Mangostin (5 μM) for 24 h was evaluated using the MTT method. Data represent the mean ± standard deviation of three independent experiments in triplicate. Data were analyzed using one-way ANOVA test followed by Dunnett post hoc test or Mann–Whitney test. Statistically significant differences are denoted as follows: * *p* < 0.05; ** *p* < 0.01; *** *p* < 0.001; **** *p* < 0.0001; and ns: non-significant.

**Figure 6 molecules-27-07362-f006:**
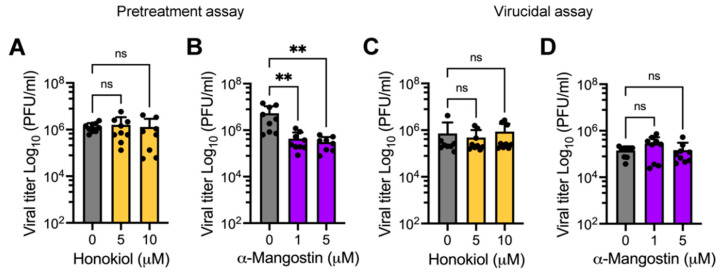
α-Mangostin pretreatment reduces MAYV progeny production. HDFs were pretreated with increasing doses of Honokiol (**A**) or α-Mangostin (**B**) for 2 h; after that, the compounds were removed, and the cells were infected with the MAYV AVR0565 strain. Following 1 h of virus adsorption, a fresh medium without the compounds was added to the cells, and they were incubated for 24 h. Next, viral titers were quantified as previously described. Following this, a 1 × 10^5^ UFP amount of the MAYV AVR0565 strain was incubated at 37 °C with the indicated concentration of Honokiol (**C**) or α-Mangostin (**D**) for 2 h. Then, the remaining virus in each experimental condition was directly calculated using plaque-forming assays. Data represent the mean ± standard deviation of three independent experiments in triplicate. Data were analyzed using one-way ANOVA test followed by Dunnett post hoc test. Statistically significant differences are denoted as follows: ** *p* < 0.01 and ns: non-significant.

**Figure 7 molecules-27-07362-f007:**
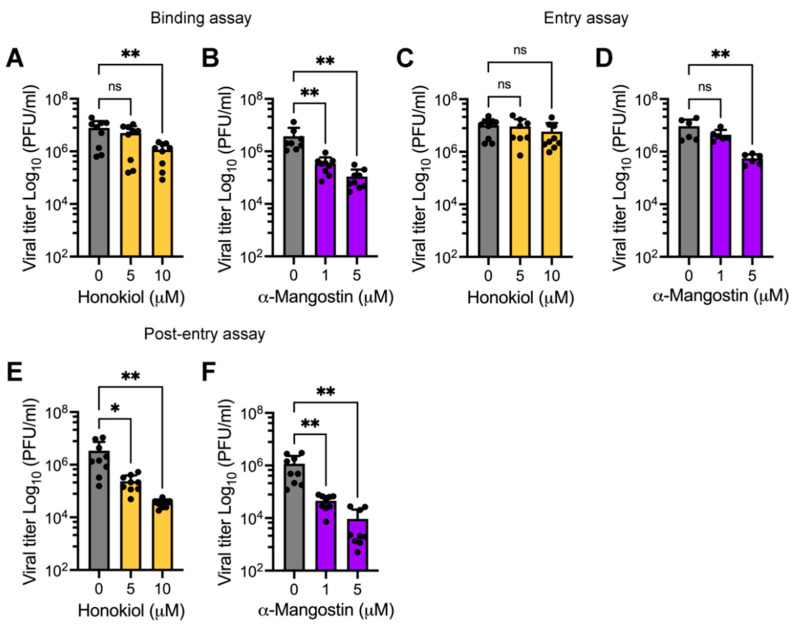
Honokiol and α-Mangostin inhibit MAYV infection at different stages of the viral life cycle. HDFs infected with MAYV AVR0565 strain at an MOI 1 and the effect of Honokiol or α-Mangostin were assessed using binding (**A**,**B**), entry (**C**,**D**), and post-entry assays (**E**,**F**). Then, viral titers were quantified using a plaque-forming assay. Data represent the mean ± standard deviation of three independent experiments in triplicate. Data were analyzed using one-way ANOVA test followed by Dunnett post hoc test. Statistically significant differences are denoted as follows: * *p* < 0.5; ** *p* < 0.01; and ns: non-significant.

**Figure 8 molecules-27-07362-f008:**
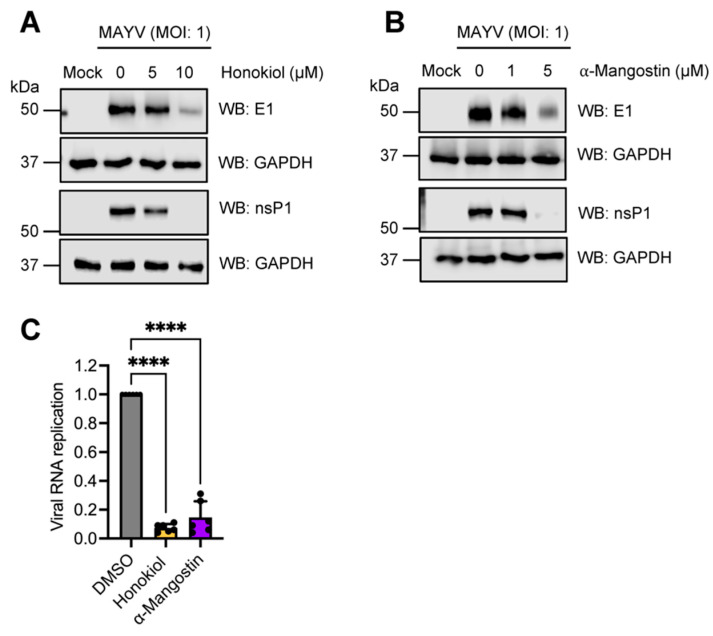
Honokiol and α-Mangostin reduce the expression of MAYV E1 and nsP1 proteins and promote a decrease in viral RNA. HeLa cells (**A**,**B**) were infected with the MAYV AVR0565 strain at an MOI of 1 and then treated with Honokiol or α-Mangostin at the indicated doses. After 24 h of incubation, protein extracts were obtained, and E1 and nsP1 viral protein levels were analyzed using Western blot. GAPDH protein was used as a loading control. Please note, kDa: kilodaltons and WB: Western blot. (**C**) HeLa cells were treated or untreated with Honokiol (10 μM) or α-Mangostin (5 μM) and viral RNA replication was assessed using RT-PCR. Data represent the mean ± standard deviation of two independent experiments in triplicate. Data were analyzed using one-way ANOVA test followed by Dunnett post hoc test. Statistically significant differences are denoted as follows: **** *p* < 0.0001.

**Figure 9 molecules-27-07362-f009:**
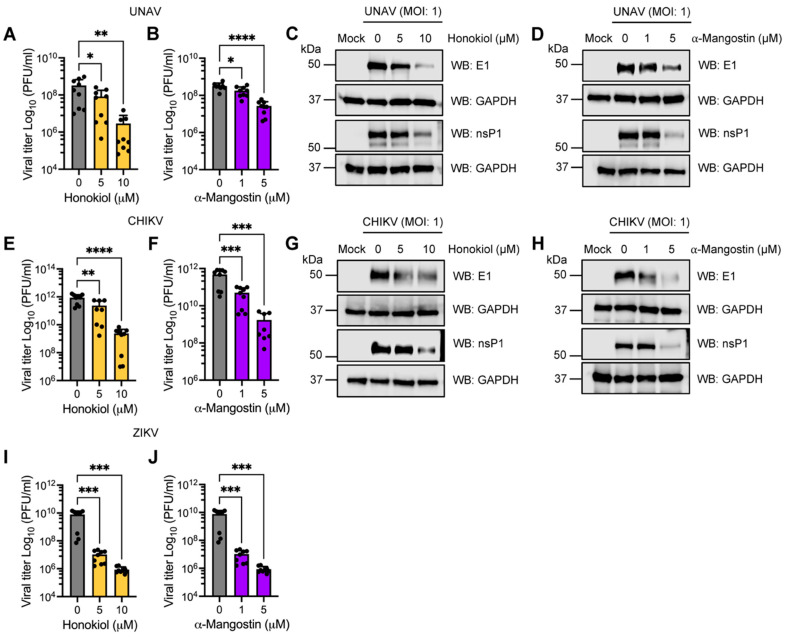
Honokiol and α-Mangostin impair UNAV, CHIKV, and ZIKV replication. Vero-E6 cells were infected with UNAV (**A**–**D**), CHIKV (**E**–**H**), or ZIKV (**I**,**J**) at an MOI of 1. After viral adsorption, increasing doses of Honokiol or α-Mangostin were added to the cells, and they were incubated for 24 h. Next, viral titers and E1 and nsP1 protein expression were evaluated using a plaque-forming assay or Western blot, respectively. GAPDH protein was used as a loading control. Please note, kDa: Kilodaltons and WB: Western blot. Data represent the mean ± standard deviation of three independent experiments in triplicate. Data were analyzed using one-way ANOVA test followed by Dunnett post hoc test. Statistically significant differences are denoted as follows: * *p* < 0.5; ** *p* < 0.01; *** *p* < 0.001; **** *p* < 0.0001; and ns: non-significant.

**Figure 10 molecules-27-07362-f010:**
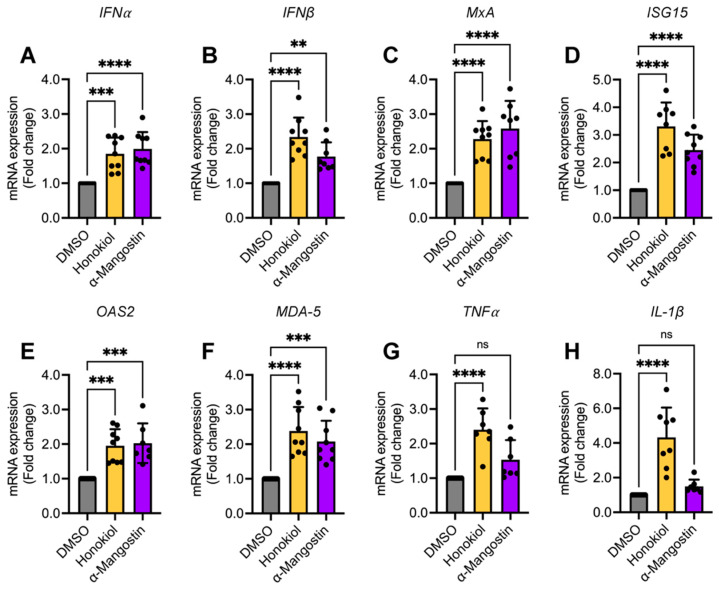
Honokiol and α-Mangostin treatment stimulate the expression of type I IFN and specific IFN-stimulated genes. HeLa cells were treated with Honokiol (10 μM) or α-Mangostin (5 μM) for 24 h. Then, total RNA was extracted and the levels of the indicated immune response genes (**A**–**H**) were assessed using quantitative RT-PCR. Relative mRNA expression in Honokiol- or α-Mangostin-treated cells was represented as fold changes as compared to DMSO-treated cells. Data represent the mean ± standard deviation of three independent experiments in triplicate. Data were analyzed using one-way ANOVA test followed by Dunnett post hoc test. Statistically significant differences are denoted as follows: ** *p* < 0.01; *** *p* < 0.001; **** *p* < 0.0001; and ns: non-significant.

**Table 1 molecules-27-07362-t001:** Primers used in this study.

Gene	Primer Sequences (5′-3′)	References
*IFN* *⍺*	Forward: GCCTCGCCCTTTGCTTTACT	[74]
	Reverse: CTGTGGGTCTCAGGGAGATCA	
*IFNβ*	Forward: ATGACCAACAAGTGTCTCCTCC	[74]
	Reverse: GCTCATGGAAAGAGCTGTAGTG	
*MxA*	Forward: GGTGGTGGTCCCCAGTAATG	[75]
	Reverse: ACCACGTCCACAACCTTGTCT	
*ISG15*	Forward: GAGAGGCAGCGAACTCATCT	[76]
	Reverse: CTTCAGCTCTGACACCGACA	
*OAS2*	Forward: AAACCAGGCCTGTGATCTTG	[77]
	Reverse: GGGCTATTTCCAGACAACGC	
*MDA-5*	Forward: GCCATTGCAGATGCAACCAG	[77]
	Reverse: TTGCGATTTCCTTCTTTTGCAG	
*TNF* *⍺*	Forward: CAGAGGGAAGAGTTCCCCAGGGACC	[74]
	Reverse: CCTTGGTCTGGTAGGAGACGG	
*IL-1β*	Forward: AACCTCTTCGAGGCACAAGG	[77]
	Reverse: GTCCTGGAAGGAGCACTTCAT	
*β*-actin	Forward: AGAGCTACGAGCTGCCTGAC	[78]
	Reverse: AGCACTGTGTTGGCGTACAG	
MAYV	Forward: CATGGCCTACCTGTGGGATAATA	
	Reverse: GCACTCCCGACGCTCACTG	

## Data Availability

Not applicable.

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
