# Peer review of "Honokiol and Alpha-Mangostin Inhibit Mayaro Virus Replication through Different Mechanisms"

_molecules, 2022, doi:10.3390/molecules27217362_

Round 1

Reviewer 1 Report

The study of Valdes-Torres & Campos et al. characterizes the natural compounds Honokiol and α-Mangostin as inhibitors of authentic MAYV infection.  Vero-E6, HDFs and HeLa cells were infected in a range of in vitro assays to determine the likely mode of action for these compounds. Viral inhibition by these compounds was also observed against diverse arboviruses including UNAV, CHIKV and ZIKV, suggesting broad spectrum activity. While subtle differences were observed between the two compounds, they appear to target multiple stages of the viral life-cycle, including binding and entry, but do not act directly on virions in a virucidal assay. Post-entry treatment of cells was the most effective method to inhibit viral infection. The authors demonstrate that both compounds can upregulate the  expression of IFN, IFN-stimulated genes and proinflammatory cytokines in HeLa cells.  Consequently the authors postulate that these compounds prime cellular antiviral responses via targeting the IFN pathway.

In general, the manuscript is nicely written, data is clearly presented and experiments are correctly designed. I have some issues with the data interpretation, specifically concerning the use of specific cell-lines for experiments, which should be addressed before acceptance.

The authors conclude the one of the major mechanisms by which the compounds act is by activation of cellular antiviral responses, which would fit with the efficient inhibition they see when the compounds are added at the post-entry step. The authors use Vero-E6, HDFs and HeLa cells in their infection experiments to characterize the compounds mechanism of action. VeroE6 cells do not produce endogenous IFN upon stimulation, although they can respond to exogenous IFN treatment. However, this can differ between different batches of VeroE6 cells. Vero-E6 also possess low baseline ISG expression. Thus, the potent antiviral effects observed in VeroE6 cells in Figure 3, 4 and 9 may be independent of the IFN system which primes cell-intrinsic antiviral responses? Also, if the compounds activate the IFN system, then pre-treatment of HDFs in Figure 6 should also inhibit replication, but no inhibition is observed for Honokiol? The authors should further characterize the ability of the different cell-lines used to produce IFN and up-regulate IFN-stimulated genes, and modify the conclusions for each figure accordingly. PolyI:C stimulation (dsRNA mimic) of the x3 cell-lines followed by qPCR of ISGs at 24 hours post transfection would be an appropriate control experiment as there is no chance of viral antagonism here. In my opinion, the antiviral effects observed in VeroE6 are unlikely to be mediated by the IFN system. This should be acknowledged, or confirmed experimentally.

The potent inhibition seen in VeroE6 cells occurs at a post entry step, as the compounds are added after infection. These stages could include uncoating, translation, replication, assembly or release. As reduction in E1 expression is observed by IF (Figure 4) and E1/nsP1 is reduced in Western blot (Figure 9), translation and/or replication are likely targeted, rather than virion assembly or release. At the very least the authors should discuss this. Ideally experiments should be designed to pinpoint the post entry life-cycle step at which these compounds act.

One final request. The authors indicate the compounds stimulate the induction of the IFN system the mediate their antiviral effects. This is shown in only in HeLa cells. Do Honokiol and α-Mangostin possess similar structures to any PAMP danger signals which activate the IFN system (eg dsRNAs, etc)? This should be discussed.

Author Response

Dear Reviewer,

In the attached document you will find our responses.

Thank you very much for your valuable feedback.

Best regards,

José González Santamaría.

Reviewer 2 Report

In the manuscript “Honokiol and alfa-Mangostin inhibit Mayaro Virus replication by stimulating the type I Interferon pathway” the authors describe the antiviral abilities of several natural compounds to determined their potential as therapeutic agents for arboviruses. The paper is well written and presented. The findings are of significance to the arbovirus community as there are not any available antiviral compounds commercially available for treatment. A few minor suggestions for improvement:

Title: alfa-Mangostin should be “alpha- Mangostin”

Line 37: Alfavirus should be “Alphavirus”

Figure 4, the IFA results are difficult to see. One would expect to see more staining for MAYV (is that the red stain?) with no treatment. As presented the colors are difficult to see and it looks like the infection control was poor. Perhaps a zoomed in image and more details on the color of the staining would be helpful.

Lines 311-316: it would be helpful to the audience to state the Zika virus is a Flavivirus and does not posses E1 or nsP1, the lack of WB data for Zika would be odd to a broad readership. A WB for Zika proteins would be a great benefit to the paper.  

Author Response

Dear Reviewer,

Thank you very much for your valuable feedback.

Best reagards,

José González Santamaría.

Round 2

Reviewer 1 Report

The revised manuscript can be accepted. The authors have addressed the highlighted issues.